# Thermal and Nutritional Strategies for Managing *Tenacibaculum maritimum* in Aquaculture: A Welfare-Oriented Review

**DOI:** 10.3390/ani15172581

**Published:** 2025-09-02

**Authors:** Raquel Carrilho, Márcio Moreira, Ana Paula Farinha, Denise Schrama, Florbela Soares, Pedro Rodrigues, Marco Cerqueira

**Affiliations:** 1Centro de Ciências do Mar do Algarve (CCMAR/CIMAR LA), Campus de Gambelas, Universidade do Algarve, 8005-139 Faro, Portugal; rvcarrilho@ualg.pt (R.C.); macerqueira@ualg.pt (M.C.); 2Campus de Gambelas, Universidade do Algarve, 8005-139 Faro, Portugal; marciojvmoreira@gmail.com; 3Bio1OneHealth, Lda., Campus de Gambelas, Universidade do Algarve, Pavilhão B1, 8005-139 Faro, Portugal; denise.bio1onehealth@gmail.com; 4Escola Superior Agrária de Santarém, Instituto Politécnico de Santarém, Quinta do Galinheiro-S. Pedro, 2001-904 Santarém, Portugal; ana.resende@esa.ipsantarem.pt; 5IPMA–EPPO, Portuguese Institute for the Sea and Atmosphere, Aquaculture Research Station, 8700-194 Olhão, Portugal; fsoares@ipma.pt; 6S2AQUA—Collaborative Laboratory, Association for a Sustainable and Smart Aquaculture, 8700-194 Olhão, Portugal

**Keywords:** tenacibaculosis, thermal therapy, behavioural fever, functional feeds, immunostimulants, fish welfare

## Abstract

Maintaining fish health in aquaculture systems remains challenging due to pathogenic infections causing economic losses. Conventional disease management relies on antibiotics and chemical treatments, raising environmental and public health concerns. This review explores fish-friendly alternatives using temperature modulation and dietary interventions to enhance immune responses. It also examines the potential benefits of integrating these approaches to promote sustainable and effective disease prevention in aquaculture.

## 1. Introduction

Aquaculture is one of the fastest-growing food production sectors. However, its tendency towards intensification can lead to poor fish welfare and environmental challenges [1]. Farmed fish are frequently subjected to several stressors that impair their innate immune defences and increase their susceptibility to diseases [2,3,4]. These stressors include high stocking densities, hypoxia, handling, temperature fluctuations, and low water quality, among others [5]. Moreover, climate change, which includes rising temperatures, shifting precipitation patterns, and ocean acidification, poses a significant risk to the sector since it can facilitate pathogen proliferation and increase incidence period and disease transmission among fish populations [6,7]. While high temperatures may temporarily boost fish immune function, prolonged exposure or fluctuating temperatures often induce stress, impairing immunity and increasing disease susceptibility [7]. Another concern about climate change is the potential emergence of pathogens that were restricted by environmental conditions [7]. Additionally, some pathogens are opportunistic, persisting in the environment or acting as asymptomatic carriers in fish. This makes aquaculture facilities highly vulnerable to disease outbreaks, thereby hindering the development of efficient, economical, and sustainable aquaculture systems [8].

Several bacterial diseases afflict cultured fish, including vibriosis [9], furunculosis [10], streptococcosis [11], edwardsiellosis [12,13], and tenacibaculosis [14]. Tenacibaculosis is a disease with significant negative economic impact caused by *Tenacibaculum* species (family *Flavobacteriaceae*, phylum Bacteroidota) [14]. This bacterial disease, primarily caused by *Tenacibaculum maritimum*, is associated with mortalities in various commercially valuable marine species worldwide and can be found in the environment, associated with marine organisms or organic matter [14,15,16]. Other names have been used to designate this disease, including eroded mouth syndrome, black patch necrosis, and gliding bacterial diseases of sea fish [17].

Tenacibaculosis is characterised by gross lesions on the body surface, such as ulcers, necrosis, tail rot, eroded mouth, frayed fins, and occasionally gill and eye necrosis [16]. In aquaculture systems, antimicrobial agents are widely used to combat *T. maritimum* infections; however, the selection of the therapeutic compound and its application protocols differ according to country/region regulations [18]. The repeated use of antibiotics promotes the emergence of antimicrobial-resistant (AMR) bacteria and has harmful effects on both the environment and consumer [19]. Moreover, there is a lack of effective vaccines against this bacterium, with only one commercially available option, ICTHIOVAC^®^-TM (HIPRA, Amer, Spain), which is species-specific for turbot (*Scophthalmus maximus*) and provides only short-term immunity lasting six months [20]. This limited duration requires further vaccination, increasing production costs and fish handling stress while providing inadequate protection during extended production cycles. Thus, the research and implementation of novel strategies to mitigate and/or prevent infections by this bacterium are imperative for transforming aquaculture disease management from reactive pathogen suppression to proactive host resilience enhancement.

This narrative review presents our critical evaluation and perspective on welfare-oriented alternatives to conventional *T. maritimum* management in aquaculture, with the aim of stimulating scientific discourse on this unexplored topic. We evaluate thermal interventions that exploit natural behavioural fever responses and nutritional approaches using functional feeds, assessing their individual theoretical potential and empirical evidence. Importantly, we distinguish between established findings from controlled studies and our proposed integration hypothesis, which remains untested and requires rigorous validation before commercial implementation.

### Methodological Approach

This review adopts a critical framework that distinguishes between empirically validated strategies and theoretical integration concepts (Figure 1). Individual thermal interventions have been demonstrated to be effective in controlled laboratory studies across multiple fish species, with behavioural fever responses documented in zebrafish [21], Atlantic salmon [22], and Nile tilapia [23]. Similarly, nutritional interventions using specific functional feed additives have shown measurable immune enhancement and pathogen resistance in controlled trials [24,25,26]. However, the integration of thermal and nutritional strategies represents an untested hypothesis, lacking empirical validation. No published studies have investigated combined thermal–nutritional protocols against *T. maritimum*. This integration concept should be, therefore, understood as a research framework requiring rigorous testing rather than an established management approach.

We conducted a comprehensive literature search across PubMed, Web of Science, and Scopus using a structured Boolean strategy. Primary search terms included the following: (“*Tenacibaculum maritimum*” OR “tenacibaculosis”) AND (“thermal therapy” OR “behavioural fever” OR “temperature management”) AND (“functional feeds” OR “immunostimulants” OR “probiotics”) AND (“aquaculture” OR “fish farming”). Secondary searches refined the scope with targeted phrases such as “behavioural fever fish,” “thermal manipulation aquaculture,” “marine algae fish feed,” “probiotic fish disease, “welfare-oriented framework”, and “precision health management”. The inclusion criteria were as follows: (1) peer-reviewed studies addressing *T. maritimum* pathogenesis, prevention, or control; (2) research on thermal or nutritional interventions for managing fish diseases; (3) welfare-focused approaches in aquaculture; and (4) empirical studies evaluating alternative disease management strategies. We prioritised the recent literature (2015–2025) when available, particularly for emerging areas such as climate-adaptive strategies, behavioural fever, and welfare-oriented disease management. However, due to the limited number of studies in these specific domains, earlier foundational research was also included where necessary to provide conceptual grounding and context.

Given the exploratory nature of this review and the limited evidence available on the integration of welfare-oriented frameworks, particularly concerning thermal and nutritional strategies, a narrative approach was arguably the most appropriate and feasible method. This approach enabled conceptual synthesis and the identification of research priorities while inherently reflecting our scientific perspective and interpretation to stimulate broader scientific discussion.

## 2. *Tenacibaculum maritimum* Management: Limitations of Conventional Approaches

### 2.1. Understanding the Pathogen–Host Interactions

*T. maritimum* is a Gram-negative bacterium (Family *Flavobacteriaceae*) with distinct biological characteristics that determine its pathogenic potential across fish species. It thrives under conditions commonly associated with intensive aquaculture, e.g., proliferating in environments where fish are crowded, stressed, and immunocompromised, conditions that significantly compromise welfare [27,28]. The pathogen ability to form biofilms allows it to persist on surfaces and equipment, creating what essentially becomes a long-lasting reservoir of infection within production systems [29].

The management of *T. maritimum* is particularly challenging due to its antigenic diversity, with four distinct serotypes (O1–O4) exhibiting different behaviours across host species and environmental conditions [30]. Molecular typing techniques reveal an additional layer of complexity beyond serological classification, exposing significant genetic variation within serotypes that can influence virulence expression and vaccine cross-protection. Techniques such as multiplex PCR, multilocus sequence typing (MLST), and whole-genome sequencing demonstrate that phenotypically similar isolates may contain distinct virulence gene profiles, thereby complicating strain-specific management strategies [31]. Additionally, according to Escribano et al. [32], serotype O1-O3 strains have similar enzymatic profiles, being homogeneous, while some isolates from serotype O4 show heterogeneity. This antigenic diversity, coupled with the functional heterogeneity of fish immune systems, explains why a vaccine effective in turbot may fail in gilthead seabream (*Sparus aurata*) and why treatment protocols that succeed on one farm can prove ineffective on another [33]. A recent study demonstrated that serologically similar isolates of *T. maritimum* may differ significantly in their genomic profiles, including virulence-related genes, highlighting the limitations of serotyping alone in guiding vaccine strategies [31]. Moreover, the bacterium employs sophisticated virulence mechanisms, including type IX secretion systems and outer membrane vesicle (OMV) production, which facilitate tissue invasion and immune evasion [32,34].

Clinical manifestations, including mouth erosion, fin necrosis, and skin ulcerations, reflect the end stage of a complex pathological process that harms fish welfare long before external signs become visible [35]. These pathological changes not only suggest underlying pain and tissue damage but also disrupt normal behaviours like feeding, swimming, and social interaction, thereby raising further welfare concerns.

The disease progression is highly species-specific, reflecting different evolutionary adaptations and immune competence. For instance, warm-water species like Nile tilapia (*Oreochromis niloticus*) and channel catfish (*Ictalurus punctatus*) often show increased susceptibility to bacterial and parasitic infections as temperatures rise. This is especially true near the upper thermal tolerance limits of both the host and the pathogen [36]. In contrast, cold-water species such as Arctic char (*Salvelinus alpinus*) and brown trout (*Salmo trutta*) may become more vulnerable during periods of temperature fluctuation, when immune suppression or heightened pathogen virulence can compromise their defences [37]. These interspecific differences emphasise the need for species-specific disease management strategies.

Susceptibility to *T. maritimum* is not only species-dependent but also varies across life stages [19,38]. Juvenile fish face particular vulnerability due to their developing immune systems and a higher surface-area-to-volume ratio, which increases exposure to external pathogens [35]. These factors make early life stages more prone to rapid disease progression, often resulting in severe clinical signs within a short period. The limited immune capacity in juveniles also reduces the efficacy of conventional treatments, which may be too slow or stressful to offer effective intervention. This life-stage-specific sensitivity highlights the need for age-appropriate disease management strategies and underscores the urgency of developing welfare-conscious alternatives that can be applied proactively.

### 2.2. Environmental and Management Drivers of Disease Emergence

While understanding virulence and host immune responses is essential, disease expression in farmed fish is strongly modulated by husbandry conditions. In the same context, while temperature plays a key role in disease expression, fish welfare is shaped by multiple interacting factors that influence susceptibility to *T. maritimum* and recovery outcomes.

High stocking densities can elevate stress markers (e.g., cortisol), impair immune responses, and damage mucosal barriers [39]. A fish already immunocompromised by high stocking density becomes more susceptible to temperature fluctuations, thereby increasing susceptibility and infection severity [40]. In Tasmania, lower mortality rates caused by *T. maritimum* have been linked to modified management strategies, including reductions in stocking density [41]. Handling procedures such as grading and transport cause acute stress and physical injury, weakening protective barriers and extending vulnerability to infection [42]. Poor water quality, particularly low oxygen (<6 mg/L), high ammonia (>0.5 mg/L), and unstable pH levels, can further compromise gill health and immune defences, increasing infection risk even at otherwise acceptable temperatures [40]. Irregular feeding or nutritionally imbalanced diets weaken immune competence and increase disease susceptibility [43]. These interconnected welfare challenges explain why *T. maritimum* outbreaks are not strictly temperature-dependent. This cascade effect can quickly overcome conventional control strategies, particularly in large-scale operations. Effective disease prevention requires, then, the integrated management of welfare factors beyond focusing solely on temperature optimisation.

Climate change adds a layer of complexity that directly impacts intervention strategy design as reviewed by Okon et al. [7]. Since the preindustrial era, average global surface temperatures have risen by 0.85 °C, with some of the most severe forecasts estimating a further rise of up to 3.5 °C by the end of the century [44]. These increases alter thermal intervention parameters, progressively reducing available thermal windows before reaching species-specific critical maxima. Gilthead seabream exemplifies this challenge. Currently, it is produced at optimal temperatures of 18–26 °C with the critical thermal maximum at 35.5 °C [45], meaning a 9.5 °C thermal safety margin. However, projected warming scenarios reduce this margin to 5.5–7.5 °C by 2050, potentially making current thermal intervention protocols ineffective or harmful. Similar constraints affect other commercially important species, requiring specific thermal management strategies [46,47].

Pathogen ecology and virulence patterns are also shifting due to climate change. Rising baseline temperatures expand the pathogen’s optimal growth window while extending seasonal infection pressure. One example would be in the Mediterranean aquaculture regions experiencing *T. maritimum*-favourable conditions for 2–3 additional months annually compared to pre-2000 baselines [48]. Relatedly, rising temperatures are expanding *T. maritimum* geographic range while intensifying seasonal infection pressure in previously stable regions [7,15]. The optimum temperature growth range for *T. maritimum* is between 15 and 30 °C, with higher disease risks observed above 15 °C and under high salinities in the range (30–35 PSU) [16,49,50]. Both environmental parameters, increasingly altered by climate change in aquaculture systems, are recognised as key drivers of bacterial proliferation and heightening infection pressure. Within this scope and since 2006, *T. maritimum* has been identified in a growing number of host species and regions. In Europe, it has been isolated from farmed tub gurnard (*Chelidonichthys lucerna*) and wild turbot in Italy [51] and from lumpfish (*Cyclopterus lumpus*) in Norway [52]. Infections have been reported in Asian in Korean olive flounder (*Paralichthys olivaceus*) [53] and in Africa in gilthead seabream and European sea bass (*Dicentrarchus labrax*) in Egypt [54,55]. In Oceania, the same pathogen was isolated from Chinook salmon (*Oncorhynchus tshawytscha*) and orbicular batfish (*Platax orbicularis*) [56,57], indicating ongoing geographic expansion in parallel with global warming trends. Ocean acidification and changing precipitation patterns create additional stressors that may compromise fish immune function and increase susceptibility to bacterial infections. These environmental changes interact synergistically with temperature effects, potentially overwhelming immune systems already stressed by suboptimal thermal conditions.

With the above information in mind, it is projected that traditional strategies designed for stable environments are unlikely to remain effective in the changing conditions aquaculture will face in the coming decades. The evolving nature of disease dynamics under climate stress highlights the need for robust and flexible management frameworks.

Likewise, nutritional strategies may require reformulation to meet shifting metabolic demands and increased environmental stress. Most importantly, integrating thermal and nutritional approaches must account for the complex, non-linear effects of climate change, which may challenge many of the assumptions that current research relies on.

### 2.3. Limitations of Conventional Treatment Strategies

The current approach to managing *T. maritimum* infections in aquaculture is often paradoxical, involving treatments that may conflict with long-term sustainability and fish welfare goals. While antibiotics can offer short-term control of bacterial load, their administration can involve handling procedures such as netting, crowding, and confinement. These conditions are known to exacerbate stress and compromise welfare in already infected fish [19,58,59,60] and do not only impair immune responses but may also facilitate further disease transmission among susceptible individuals. Antibiotics conflicts with long-term sustainability through the following factors: (1) environmental persistence causing aquatic ecosystem disruption, (2) selection pressure driving antimicrobial resistance development, (3) bioaccumulation in sediments affecting benthic communities, and (4) consumer concerns about antibiotic residues reducing market acceptability [61,62]. Furthermore, many antibiotic efficacy studies rely on challenge models that expose fish to pathogen concentrations significantly exceeding those encountered under natural farming conditions. While these models may standardise disease induction, they risk inflating perceived treatment efficacy and obscure the risks of antimicrobial resistance (AMR) development under realistic pathogen loads. Relatedly, a growing concern is the emergence of antimicrobial resistance (AMR).

AMR is a complex issue that arises at the intersection of human, animal, and environmental health, requiring a One Health approach to fully address its implications [63]. In aquaculture, amphenicols are the most used antimicrobial class globally, particularly florfenicol in European and North American operations, followed by fluoroquinolones (primarily used in Asian aquaculture) and tetracyclines (common in Latin American and some European systems) [64]. The use of antimicrobials can contribute to the emergence of resistant bacteria and resistance genes in aquatic systems, with the potential for these to disseminate into terrestrial ecosystems [63]. Although *T. maritimum* remains largely susceptible to commonly used antibiotics such as florfenicol and oxytetracycline, regional patterns of resistance are becoming increasingly evident, raising the risk of treatment failures [65]. This creates a concerning cycle of dependency on more potent or frequent chemical interventions, which carry greater ecological risks and welfare trade-offs.

Vaccination represents perhaps the most significant limitation of conventional approaches. Despite decades of research investment, only one commercial vaccine is currently available, and specific to turbot, with some reports of autovaccines or autogenous bacterins for this disease already developed and implemented in some production facilities [14,35,66]. While outer membrane-vesicle-based vaccines have demonstrated encouraging results in laboratory trials, the gap between experimental success and real-world implementation remains a significant challenge [20]. This limited progress reflects the fundamental challenges posed by *T. maritimum* antigenic diversity and the complex species-specific immune responses required for effective protection. For instance, in Atlantic salmon (*Salmo salar*), tenacibaculosis causes yellow mouth disease, i.e., yellow plaques and ulcers in the mouth of the fish. A recent study using isolates from western Canada succeeded in formulating a vaccine against yellow mouth that triggered an antibody response, but it failed to provide protection during challenge trials [67].

A critical examination of the conventional treatment literature also reveals significant methodological limitations that may overestimate efficacy and underestimate welfare impacts. Assessments of fish welfare impacts associated with conventional treatments prioritise short-term endpoints such as survival or gross lesion reduction while failing to evaluate stress-related biomarkers, behavioural alterations, or long-term physiological consequences. Moreover, evidence suggests that pharmaceutical-industry-sponsored studies are more likely to report favourable outcomes, whereas independent research more frequently documents limited efficacy or adverse effects [68]. This distortion of the evidence base fosters a wrong perception of conventional treatment success. Many experiments are conducted under controlled laboratory conditions that do not reflect the heterogeneous realities of commercial aquaculture, limiting its validity and applicability across diverse production contexts. These knowledge gaps constrain our ability to conduct comprehensive risk–benefit analyses when comparing conventional versus alternative disease management strategies.

### 2.4. Why Alternative Approaches Are Essential

The limitations of conventional aquaculture management create compelling arguments to explore alternative strategies that align more closely with biological and welfare principles. Based on available evidence, we argue that healthy, unstressed fish with optimal immune function may resist bacterial infection even under challenging conditions, though this relationship requires further investigation in commercial settings [7]. This insight shifts the focus from pathogen elimination to host resilience enhancement, a fundamentally different approach that aligns with welfare principles.

The biological characteristics of *T. maritimum* present opportunities for innovative management strategies. Its temperature-dependent growth patterns suggest that controlled thermal manipulation could be exploited as a therapeutic tool. Furthermore, since infection often initiates through mucosal surfaces (skin, gills, intestine) [69], nutritional strategies targeting gut health and immune function could provide effective protection and contribute to improved disease resistance.

Importantly, fish have evolved innate defence mechanisms against bacterial pathogens, including behavioural fever responses [70,71,72] and mucosal immune systems. However, these natural defences are often inadvertently suppressed by contemporary aquaculture practices. By understanding and supporting these natural defences through targeted environmental and nutritional interventions, we can offer a more sustainable path toward disease control. Such approaches have the potential to not only reduce reliance on antibiotics but also improve overall fish welfare and disease resilience (Table 1).

Genetic improvement through selective breeding offers a complementary strategy to welfare-oriented approaches in disease management. Resistance to bacterial diseases in aquaculture exhibits moderate heritability, typically between 0.15 and 0.40, indicating substantial potential for genetic improvement (e.g., survival against *Streptococcus agalactiae* in Nile tilapia shows h^2^ ≈ 0.15–0.38; *Flavobacterium columnare* resistance h^2^ ≈ 0.15) [77,78]. In European seabass and gilthead seabream, studies report heritability values in the range 0.24–0.43 for resistance to Viral Nervous Necrosis (VNN), making them suitable targets for disease-resistance breeding [78]. Selective breeding in seabass has demonstrated 30–80% family-level differences in survival under bacterial challenge, while genomic selection techniques accelerate progress by ~20–30% compared to traditional pedigree selection by improving prediction accuracy by 13–27% [78]. However, long generation intervals (typically 2–4 years) and potential trade-offs with traits such as growth rate or feed conversion efficiency require specific program design. In this framework, genetic improvement could be integrated with welfare-enhancing interventions, such as thermal and nutritional strategies, to enhance host resistance holistically and reduce reliance on chemical treatments. We present these arguments as our perspective on the potential for alternative approaches, recognising that comprehensive validation studies are still needed to confirm these benefits in practice.

### 2.5. Implementation Barriers and Commercial Reality

Translating successful welfare-oriented alternatives from laboratory settings to commercial reality has substantial barriers across multiple domains, and these are often underestimated in the research literature. These barriers include economic, regulatory, infrastructure, and operational domains, each posing specific obstacles to large-scale adoption. Economic limitations are among the most significant. Thermal intervention systems require major capital investments and ongoing energy costs that can compromise financial viability, and constraints vary depending on the production system. Recirculating aquaculture systems (RASs) offer greater control over environmental conditions but often require substantial investment to allow thermal regulation. Electricity is commonly reported as the second largest operational expense, accounting for approximately 7% of total production costs [79]. While specific data on the additional energy costs of thermal interventions are lacking, it is likely that sustained heating or cooling to induce behavioural fever would further increase operational energy demands. Moreover, upgrading mid-scale RAS units (1000–3000 m^3^) with thermal control systems may require substantial capital investment, depending on system design and regional climate conditions. In contrast, sea cage operations face challenges in manipulating temperature, as they rely on natural water conditions. However, some vertical temperature gradients (1–2 °C) do exist, and strategies like raising submerged cages toward the surface may offer limited thermal modulation. While not precise, these approaches suggest that behavioural fever strategies in cages are not impossible but are difficult to implement consistently with current technology.

Regulatory obstacles add another layer of difficulty, particularly for functional feed additives. Many additives operate in a regulatory grey zone, lacking clear approval or prohibition. In the EU, for instance, EFSA approval for novel feed ingredients requires expensive safety dossiers and timelines of 2.6 ± 1.2 years [80]. In the United States, FDA requirements for efficacy data pose similar challenges for aquaculture additives, a requirement many probiotic and phytochemical suppliers struggle to meet. This regulatory uncertainty discourages innovation and creates liability concerns for producers. Furthermore, most aquaculture farms are not equipped with the environmental control systems needed for implementing behavioural fever protocols, making infrastructure limitations an additional constraint. Upgrading the infrastructure may involve installing automated temperature control, continuous monitoring, and safety systems to prevent thermal shock. Complementarily, functional feeds often require cold storage and specialised handling equipment, adding logistical complexity and cost for farmers.

Quality control and product consistency are also challenging. Unlike pharmaceutical products, aquaculture probiotics and additives face minimal oversight regarding viable cell counts, strain identification, and shelf stability [81]. This lack of standardisation leads to variable performance in the field, reducing producer trust in such products. Other barriers to implementation are the operational complexity associated and production scale. For instance, thermal strategies require skilled staff capable of interpreting environmental data and adjusting systems in real time. Similarly, functional feed protocols demand careful timing, proper storage, and whole integration with feeding routines. These demands often exceed the technical and human capacity of small-to-medium farms, which represent a large part of global industry. Additionally, while large operations may afford these innovations, smaller producers often face disproportionate costs per unit, and, therefore, sustainable alternatives are of difficult access. This dynamic limits the broader transformation of the sector.

## 3. Thermal Strategies for Managing *T. maritimum* Infections in Fish

### 3.1. Temperature as a Driver of T. maritimum Virulence and Host Susceptibility

Temperature plays a fundamental role in regulating both fish physiology and bacterial pathogenicity. In teleosts, water temperature modulates key biological processes such as growth [82,83], immunity [84], reproduction [85], basal metabolic processes [86], and survival [87,88]. Within their thermal tolerance window, fish may benefit from increased metabolic activity and feed conversion efficiency. For instance, common carp living freely in a large pond have been observed warming their bodies by up to 4 °C above the surrounding water by spending time in sunny areas. The more time they spend in these areas, the faster they grow [89]. However, temperatures that exceed the species-specific optimal range can induce stress responses, including activation of heat shock proteins (HSPs), oxidative stress, and, ultimately, cell death or increased mortality [90].

The virulence of *T. maritimum* is also strongly temperature-dependent. The bacterium exhibits optimal growth at approximately 30 °C and remains viable within a range of 15–34 °C [16]. Below 15 °C, bacterial growth and biofilm formation are significantly reduced. Temperature thresholds for disease outbreaks vary by host species (Table 2). Accordingly, wedge sole outbreaks occurred at 20.5 ± 1.5 °C [91]; Japanese flounder (*Paralichthys olivaceus*) showed peak mortality between 17 °C and 26 °C [92]; and European seabass demonstrated higher *T. maritimum* prevalence during summer outbreaks compared to winter outbreaks [54].

In cold-water species such as rainbow trout (*Oncorhynchus mykiss*), Rebl et al. [95] reported a decline in innate immune function under elevated temperature conditions, resulting in increased susceptibility to bacterial infections. This increased susceptibility in these species, as for warm-water fish, results from compromised immune function when temperatures exceed their thermal optima, creating a vulnerability window that *T. maritimum* exploits. As a supportive fact, in Chilean farms, this bacterium caused severe outbreaks in rainbow trout farmed with 16.2 °C of average water temperature [94]. In the same scope, the bacterium was shown to remain pathogenic at lower temperatures. Experimental injection of extracellular products (ECPs) from *T. maritimum* into Atlantic salmon at 15–16 °C resulted in tissue necrosis and mortality [93]. These findings also suggest that *T. maritimum* is already established in cooler regions, and even modest temperature increases could facilitate further geographic expansion [15]. Table 2 describes the thermal sensitivity of *T. maritimum* across several species, underscoring the need for species-specific temperature management strategies to mitigate *T. maritimum* outbreaks effectively.

### 3.2. Behavioural Fever in Fish: Mechanisms, Evidence, and Limitations

To facilitate recovery from an infection, endotherms express physiological fever, while ectotherms, such as fish, respond behaviourally by moving temporarily to places with higher water temperatures [96]. When fish quickly start seeking out warmer water after sensing an infection, this response is known as “behavioural fever”. Fish exhibiting behavioural fever responses to pathogen challenge involve the active selection of warmer temperatures, typically 2–5 °C above baseline, which enhance immune function through accelerated cellular processes and direct pathogen suppression [21,71]. The tendency of fish to seek out warmer water when ill is not only of scientific interest but also holds practical significance for aquaculture management. If farmed fish had access to a range of water temperatures, they could better manage their health, potentially speeding up recovery and reducing the need for medications. This adaptive response is known to operate through conserved molecular pathways involving pro-inflammatory cytokines, including tumour necrosis factor-α, interleukin-6, interleukin-1β, and production of prostaglandin E2, which trigger reprogramming of thermosensory systems by downregulating TRPV4 channel expression (Figure 2) [22].

It is essential to understand that behavioural fever differs from what is sometimes referred to as emotional fever or stress-induced hyperthermia, where fish prefer warmer temperatures to cope with stress. Both are types of behavioural thermoregulation, but whether fish truly show a temperature rise due to stress is still debated [97,98,99] and requires further research. For instance, initial findings of emotional fever in zebrafish (*Danio rerio*) [100] were not replicated in subsequent studies, suggesting methodological sensitivity [99]. Behavioural fever caused by infection is scientifically supported, as demonstrated by the evidence below. However, distinguishing between true behavioural fever (adaptive immune response to infection) and stress-induced “emotional fever: (non-adaptive temperature preference under psychological stress) is complex because both involve similar thermosensory pathways and temperature-seeking behaviours yet have opposite welfare implications, one being beneficial and the other indicating distress.

Zebrafish infected with spring viraemia virus exhibited no clinical signs of disease and experienced no mortality when permitted to freely navigate a temperature gradient. These fish achieved full recovery and were cleared of the virus within a week. Researchers observed a robust activation of antiviral gene expression in fish permitted to thermoregulate freely, a response absent in the control group maintained at a constant temperature [21]. Similar outcomes were seen in common carp infected with Cyprinid herpesvirus 3. Carp that could choose between tanks with progressively higher temperatures (24 °C, 28 °C, and 32 °C) showed increased survival to infection. The “fever” response was linked to the activation of immune signals called inflammatory cytokines [101].

In rainbow trout, exposure to bacterial LPS resulted in elevated IL-1β expression and a behavioural preference for warmer water [72]. Similarly, Atlantic salmon demonstrated enhanced immune responses when permitted to thermoregulate during infection with infectious pancreatic necrosis virus (IPNV) [22]. Nile tilapia also showed a preference for warmer water after being infected with the bacterium *Streptococcus iniae*, indicating that this self-heating behaviour may be common across different species [23]. These findings suggest that allowing fish to access warmer water during illness can significantly enhance recovery outcomes. However, this phenomenon remains inconsistently demonstrated across species (Table 3).

#### Limitations of Behavioural Fever

Important limitations of behavioural fever research need to be acknowledged. Most of what we know comes from laboratory studies using thermal gradient systems that differ greatly from real aquaculture environments [21,102,103]. In production settings, fish are exposed to multiple stressors including crowding, handling, fluctuating water quality, and social hierarchies, all of which may suppress, delay, or prevent behavioural fever responses [104]. For instance, in commercial ponds or sea cages, fish are subjected to fluctuating temperatures that may exceed behavioural fever ranges, variable oxygen levels that compromise immune function and increase *T. maritimum* susceptibility, social interactions creating stress hierarchies that suppress behavioural fever responses, and numerous other stressors that collectively may prevent the expression of beneficial thermoregulatory behaviours observed in laboratory studies. Additionally, the metabolic costs of maintaining an elevated body temperature are not well understood. Laboratory studies typically measure only pathogen resistance and survival, ignoring potential trade-offs including reduced growth rates, compromised feed conversion efficiency, increased energy expenditure, and elevated cortisol levels [105]. These costs may offset any immune benefits, particularly under the nutritional and space constraints from typical commercial aquaculture systems [104]. Species-specific variability in behavioural fever responses presents additional challenges. While some studies report robust responses in zebrafish and salmon, others document absent or inconsistent responses in commercially important species [106]. This variability suggests that behavioural fever may be limited to specific phylogenetic groups or environmental conditions, constraining its broad applicability across diverse aquaculture species.

The assumption that temperature preferences observed in laboratory settings also promote benefits in commercial systems remains largely untested. No studies have demonstrated that providing thermal choice options reduces mortality or improves welfare outcomes in production-scale systems infected with *T. maritimum*. This is somewhat surprising, considering the growing importance of this pathogen in both marine and freshwater environments, alongside the well-documented success of thermal preference shifts in helping fish recover from other diseases. It is plausible to think that *T. maritimum* develops in warm conditions and that infected fish will attempt to seek warmer water as a coping mechanism response if a gradient is available. Nevertheless, the gap between laboratory proof of concept and commercial validation represents a critical knowledge deficit that limits responsible implementation recommendations. Creating thermal gradients in sea cages remains technically challenging, limiting applications to land-based systems. Even in controlled environments, maintaining stable thermal gradients requires sophisticated heating/cooling systems and continuous monitoring that may exceed the technical capacity and economic resources of many aquaculture operations [107]. Addressing this gap could substantially enhance fish welfare in aquaculture by determining whether behavioural thermoregulation constitutes a viable and natural coping strategy against tenacibaculosis. It would also help clarify how environmental enrichment and system design might be adapted to support immune function not just in theory but in real farming conditions.

**Table 3 animals-15-02581-t003:** Fish-preferred temperatures during behavioural fever following pathogen infection.

Species	Pathogen	Control/Optimal Temperature (°C)	Preferred Temperature (°C)	Observations	References
*Micropterus salmoides*	*Aeromonas hydrophila*	30.5	32	2.1 °C increase in preferred temperature	[70]
*Lepomis* *macrochirus*	*Aeromonas hydrophila*	30	33	2.7 °C increase in preferred temperature	[70]
*Danio rerio*	Double-stranded RNA (dsRNA)	29	33	Mean 3 ± 0.5 °C shift in thermal preference maintained over at least 24 h; under viral infection, mortality decreased	[21]
*Oreochromis* *niloticus*	*Streptococcus iniae*	29–31	32–33	Reactive and proactive fish chose higher temperatures than naïve fish and the peak thermal response occurred at 24 h post-infection	[23]
*Oreochromis* *niloticus*	*Edwardsiella piscicida*	28	34	Reduced mortality; higher expression of genes involved in fever induction and response (e.g., TNF-α, IL-6, IL-1β, IL-8, COX-2)	[108]
*Danio rerio*	Double-stranded RNA (dsRNA)	28	32	Behavioural fever was shown only by larvae at 18–20 dpf	[109]
*Cyprinus carpio*	Cyprinid herpesvirus 3 (CyHV-3)	24–28	32	No mortality when the expression of behavioural fever was allowed	[101]
*Salmo salar*	Infectious pancreatic necrosis virus (IPNV)	15	18–20	Promotes the synthesis of pro-inflammatory cytokines; TRPV1 and TRPV4 channels coordinate temperature sensing during behavioural fever	[22]
*Salmo salar*	Infectious pancreatic necrosis virus (IPNV)	15	18	Fever in fish triggers neuro-immune interactions that modulate inflammatory response during pathogenic infection	[102]
*Oncorhynchus mykiss*	Bacterial lipopolysaccharide (LPS)	13.5	16	Increased expression of cytokine interleukin-1β	[72]

### 3.3. Practical Implementation of Thermal Therapies in Aquaculture

Artificial thermal manipulation, designed to mimic the benefits of behavioural fever, as stated, could provide the aquaculture industry with access to a non-antibiotic management strategy, i.e., using controlled, short-duration increases in water temperature to enhance host immune function while simultaneously creating suboptimal conditions for pathogen proliferation.

Some successful field applications demonstrate the practical potential of this approach. In barramundi farms, short-term heat treatments reduced mortality and accelerated recovery from viral and secondary bacterial infections, partly through heat shock protein activation [110]. Similarly, in Israeli common carp (*Cyprinus carpio*) farms, transferring fish infected with carp nephritis and gill necrosis virus (CNGV) from permissive (18–25 °C) to non-permissive (30 °C) temperatures decreased mortalities from 80–90% to approximately 40% [111]. As stated elsewhere, holding farmed Asian seabass (*Lates calcarifer*) infected with a specific viral disease at higher temperatures reduced disease prevalence and mortality levels [112]. In the same review, the authors reported on a recent project in Egypt that investigated this concept in Nile tilapia by modifying pond infrastructure to establish thermal gradients. To our knowledge, the data are still not publicly available; however, the authors stated that some farmers already believe these changes have improved survival and production. Based on the current understanding, this approach may be particularly applicable in controlled settings such as recirculating aquaculture systems (RASs) and hatcheries, where environmental parameters can be precisely managed. In such systems, implementing and fine-tuning thermal gradients or raising water temperatures as needed is possible and likely supports behavioural fever. As suggested by Grans et al. [72], these systems offer a high degree of environmental control, which can be leveraged not only for optimal growth conditions but also for improving disease resistance through temperature-based management strategies.

However, it is important to acknowledge that the outcomes of thermal interventions are likely to differ significantly across different species–pathogen combinations, highlighting the need for species-specific protocol optimisation. One example is Atlantic cod (*Gadus morhua*) infected with *Brucella pinnipedialis*, where a temperature rise from 6 °C to 15 °C successfully cleared infections but at the same time increased overall mortality compared to lower-temperature treatments [113]. This contradiction reflects the complex trade-offs between enhanced immune function and increased metabolic costs in cold-water species. At elevated temperatures, Atlantic cod experienced heightened energy demands for maintaining physiological homeostasis, simultaneously upregulating both immune responses and anti-inflammatory processes to prevent tissue damage from excessive immune activation. In another study with cod, Claireaux et al. [114] observed an increase in metabolic rate at 10 °C compared to fish at 2 °C [115]. These findings emphasise the importance of aligning thermal treatments with species-specific physiological thresholds to avoid unintended stress. For cold-water species like cod, even moderate temperature increases can exceed optimal thermal windows, creating energetic stress that reduces the benefits of the immune system [113,116].

Despite these recognised challenges, temperature interventions could be optimised for practical use in aquaculture by, for example, implementing them during high-risk periods (e.g., post-transport) and adjusting temperature increments based on fish thermal tolerance and immune response thresholds. Furthermore, to maximise therapeutic benefits while minimising potential risks, thermal strategies should be incorporated into integrated health management approaches, such as combining them with non-antibiotic interventions, including probiotics, immunostimulants, and welfare-enhancing practices, to boost disease resistance. For the management of *T. maritimum*, thermal intervention protocols appear especially promising due to the pathogen’s well-characterised sensitivity to temperature. With optimal growth at 30 °C and significant growth reduction above this threshold, controlled temperature elevations to 32–35 °C can create dual therapeutic effects: direct bacterial growth inhibition combined with enhanced host immune function. This approach is especially relevant for warm-water species already cultured near the optimal temperature range of *T. maritimum*, where modest increases can shift the balance from pathogen-favourable to host-favourable conditions.

While operational costs related to heating the water remain a practical concern, their feasibility must be evaluated on a case-by-case basis, accounting for the aquaculture system, energy pricing, species value, and disease prevalence. Comprehensive economic evaluations are lacking and yet necessary to determine cost-effectiveness across various production settings.

## 4. Nutritional Strategies: Fortified Feeds with Natural Immunostimulants

### 4.1. The Role of Nutrition in Disease Resistance Across Fish Species

Optimal feeding protocols and balanced nutrition are fundamental to sustainable aquaculture, as dietary deficiencies not only impair growth and reproduction but also compromise immune function and resistance to stress and disease [117,118,119]. Mucosal tissues (skin, gills, and the gastrointestinal tract) serve as the first line of immune defence, and their integrity is closely tied to nutritional status [120]. Strengthening mucosal immunity may, therefore, offer a preventive strategy against tenacibaculosis and reinforce overall fish health [69]. These surfaces present epithelial cells that secrete antimicrobial molecules like lysozyme, which help eliminate invading pathogens. If this barrier is affected, additional innate responses are triggered, such as cytokine production and phagocytic activity by macrophages and neutrophils [120]. The gut also plays a central role in immune defence through its resident microbiota. A balanced microbial community supports digestion, modulates mucosal tolerance, and enhances resistance to infections [121]. In contrast, microbial imbalance (dysbiosis) can increase vulnerability to pathogens [122]. Figure 3 summarises the pathway from functional feed intake to pathogen resistance with the biological responses associated with undergoing these processes. Interestingly, microbiota diversity is influenced by water temperature, with warm-water species generally showing greater microbial richness [122]. To maintain mucosal immunity and promote beneficial gut microbiota, functional feed additives—such as prebiotics, probiotics, and phytogenics—have been increasingly used in aquaculture [123]. For example, *Lactococcus lactis* subsp. *lactis* SL242 improved interleukin expression and microbiota balance in gilthead seabream [124], while *Bacillus subtilis* AB1 enhanced innate immune parameters and pathogen resistance in rainbow trout [125]. Although we have highlighted the above examples of nutritional and microbial interventions, a comprehensive examination of each additive is beyond the scope of this review.

Fortunately, in the past years, numerous in-depth reviews have rigorously evaluated the usefulness of probiotics, prebiotics, synbiotics, phytogenics, and other functional additives in aquaculture. Some standout papers include the review of Merrifield et al. [126], outlining how probiotics enhance disease resistance, growth, and gut health in finfish and crustaceans; the work from Khanjani et al. [127] with a detailed look at the benefits of integrating probiotics and phytobiotics within biofloc systems; and the review from Yadav et al. [128], which provides an extensive overview of a wide range of feed additives, highlighting their effects on growth performance, stress tolerance, gut microbiota, and immune responses. In the same line, Bharathi et al. [129] focused their research on functional feed additives (e.g., probiotics, phytobiotics, organic acids), their impact on immune response, and their compatibility with sustainable aquaculture systems, while Hernández-Contreras et al. [130] emphasised natural, eco-friendly additives and their roles in promoting sustainability and animal welfare in aquaculture. These and other reviews provide the scientific depth necessary for anyone wishing to explore the specific mechanisms, dosages, and efficacy outcomes of each additive, serving as essential resources for those seeking comprehensive, evidence-based guidance on the practical use of functional feed additives.

### 4.2. Functional Feeds and Natural Immunostimulants: Laboratory Promise and Commercial Challenges

Functional fish feeds incorporate bioactive compounds, such as phytochemicals, probiotics, and immunostimulants, beyond essential nutrients, with the goal of enhancing immune function, disease resistance, and microbiota balance [73]. These strategies show great promise in laboratory settings and offer potential for more sustainable health management in aquaculture. However, consistent translation of these benefits to commercial conditions remains limited. For instance, while chestnut polyphenols improved survival in tilapia under controlled conditions [131], such results are not consistently replicated in commercial settings. Spirulina-based diets have shown similar promise [132], but reproducibility across systems remains a limitation.

Quality control represents the most fundamental barrier to reliable commercial application. Phytochemical extracts often show substantial batch-to-batch variability in active compound concentrations due to inconsistent raw materials, unstandardised extraction methods, and the lack of clear analytical standards [133]. Consequently, commercial products may contain insufficient active compounds to achieve therapeutic effects or excessive concentrations that trigger adverse responses. This variability undermines efficacy and complicates risk assessment. Dose–response relationships remain poorly understood for many of these additives. In Senegalese sole (*Solea senegalensis*), dietary inclusion of 1% ulvan-rich extract enhanced immune responses, while a 2% inclusion suppressed them [134]. These threshold effects highlight the need for careful formulation and mechanistic insight.

Probiotic applications face equally significant commercial challenges. While species such as Bacillus spp. and lactic acid bacteria have shown antimicrobial effects in controlled studies [135,136], their effectiveness is highly dependent on strain, environment, and farming system. Outcomes observed in one site often fail to replicate in others due to differences in water quality, management practices, or microbiota composition. This site specificity limits scalability and broader adoption. Moreover, probiotic viability is often compromised during feed processing, with pelleting temperatures (85–95 °C) destroying up to 90% of organisms. Post-processing application is necessary to preserve efficacy, but it increases costs and contamination risks [137]. Shelf-life is another concern, and many commercial probiotic feeds retain less than 15% of their labelled viable counts after six months at room temperature [138].

A common assumption in both industry and research is that natural additives are inherently safe. However, compounds like β-glucans, chitosan, and other immunostimulants can overstimulate the immune system or interfere with metabolic balance, especially at high or prolonged doses. This risk is particularly relevant for species or life stages with lower metabolic capacity or immune resilience. Such unintended effects may partly explain inconsistent field results and highlight the need for dose optimisation and species-specific validation.

Economic barriers further limit adoption. Functional feed additives are widely acknowledged to raise feed formulation costs, sometimes substantially, but the specific increase varies by product, supplier, and system. These higher costs, combined with uncertainty about field performance, make producers hesitant to invest. Regulatory hurdles also add complexity, as described in Section 2.5. Finally, the field is affected by publication bias. Studies reporting positive effects of functional feeds are more likely to be published, while negative or neutral results remain underreported. This distorts the scientific literature and may lead to inflated expectations regarding efficacy. A systematic effort to publish well-designed studies with all outcomes regardless of result is essential to develop an evidence-based understanding of functional feed performance.

### 4.3. Experimental Evidence: T. maritimum-Specific Applications and Limitations

Several studies have investigated nutritional interventions against *T. maritimum* infections in aquaculture. Although encouraging, the current evidence remains limited in scope, methodological robustness, and applicability in commercial settings (see Table 4).

Marine algae represent the most extensively studied nutritional intervention for T. maritimum management, offering multiple bioactive mechanisms that target different aspects of host–pathogen interaction. *Nannochloropsis oceanica* and *Chlorella vulgaris* demonstrate direct bactericidal effects.

Recent studies by Ferreira et al. [24,25] demonstrated the bactericidal potential (through bioactive compounds including chlorophylls, carotenoids, and polyunsaturated fatty acids that disrupt bacterial cell membrane integrity) of the microalgae *Nannochloropsis oceanica* and *Chlorella vulgaris*, as well as the macroalgae *Gracilaria gracilis* and *Ulva rigida*, against *T. maritimum* in vitro. In vivo, dietary inclusion of a 4% commercial algal blend improved resistance in European seabass. The immunomodulatory mechanisms of algal supplementation operate through multiple pathways. Eicosapentaenoic acid (EPA) and docosahexaenoic acid (DHA) from these algae modulate inflammatory responses by competing with arachidonic acid in prostaglandin synthesis, promoting anti-inflammatory mediator production while maintaining immune responsiveness [25]. β-glucans present in algal cell walls activate complement pathways and enhance phagocytic activity through interaction with pattern recognition receptors on fish macrophages.

Plant-derived compounds have also been tested with mixed outcomes. Oil extracts of purple coneflower (*Echinacea purpurea*) and oregano (*Origanum vulgare*) enhanced European seabass immunity and disease resistance against *T. maritimum* [26], while the plant extract carvacrol in combination with cymene proved effective as a tenacibaculosis treatment in surge wrasse (*Thalassoma purpureum*) [139]. However, alongside regulatory approval processes, concentration-dependent effects create narrow therapeutic windows requiring precise dosing to avoid immune suppression or tissue irritation.

Probiotic-based approaches are also emerging as potential alternatives. Probiotic mechanisms against *T. maritimum* involve competitive exclusion, immune stimulation, and direct antimicrobial production. For instance, lactic acid bacteria inhibited *T. maritimum* adhesion to turbot mucus in vitro [140], and *Bacillus* strains isolated from gilthead seabream reduced pathogen biofilm formation [135]. Bacillus species demonstrate particular promise through sporulation capability that enhances survival during feed processing and storage, addressing key commercial viability concerns. Critical knowledge gaps persist regarding strain specificity, environmental compatibility, formulation stability, and large-scale delivery systems. For example, *Lactobacillus plantarum* strains effective against *Edwardsiella tarda* [136] may lack efficacy against *T. maritimum* due to different virulence mechanisms and tissue tropism. This specificity makes necessary pathogen-specific screening and validation rather than broad-spectrum probiotic applications. To date, only a few functional feed products have reached commercial relevance. One notable example is the β-glucan product MycoFence^®^, derived from *Aspergillus niger*, which reportedly conferred protection against *T. maritimum* in Atlantic salmon [141]. Although promising, these results require cautious interpretation due to species-specific responses and the limited number of standardised large-scale trials that validate its use.

Management practices can also be a limitation and impact the effectiveness of nutritional strategies regarding *T. maritimum* immunomodulation. For example, the timing of nutritional interventions critically affects outcomes. Prophylactic administration 3–4 weeks before expected pathogen exposure allows immune system priming and beneficial microbiota establishment [142]. In the same way, responses to nutritional interventions are species-specific, reflecting differences in digestive physiology and immune system organisation. It is speculated that carnivorous species like European seabass may respond differently to plant-derived immunostimulants compared to omnivorous species due to differences in digestive enzyme profiles and gut microbiota composition.

In summary, while functional feeds and natural immunostimulants show significant potential in managing *T. maritimum* infections, the current body of evidence lacks the scientific rigour and commercial validation necessary to support widespread implementation. Most studies remain confined to laboratory conditions, with small sample sizes, short durations, and limited economic analysis. To enable practical adoption, future research should prioritise large-scale, multi-site field trials with standardised protocols, species-specific safety assessments, and mechanistic insights that support welfare-oriented interventions.

**Table 4 animals-15-02581-t004:** Potential of different diet compounds to mitigate *T. maritimum* infection in fish.

Compounds	Effect	Host	References
**Algae**			
*Nannochloropsis oceanica* *Chlorella vulgaris* *Gracilaria gracilis* *Ulva rigida* Commercial blend of these algae (Algaessence Feed™, ALGAplus Lda., Ílhavo, Portugal)	40–45% bactericidal activity against *T. maritimum*; all species were effective	In vitro	[24]
Commercial blend (Algaessence Feed™, ALGAplus Lda., Ílhavo, Portugal)	4% inclusion inhibits disease progression and reduces mortality	*Dicentrarchus labrax*	[25]
**Plants**			
*Echinacea purpurea* oil extract *Origanum vulgare* oil extract	Enhanced immunity and disease resistance against *T. maritimum*	*Dicentrarchus labrax*	[26]
Plant extract carvacrol combined with cymene	14 days of treatment resulted in no clinical signs of disease and no mortality	*Thalassoma purpureum*	[139]
**Probiotics**			
LAB	Antimicrobial activity against *T. maritimum* and inhibition of its adhesion to turbot mucus	In vitro	[140]
*Bacillus* spp.	Some fish-gut Bacillus spp. isolates and their extracellular NACs * inhibited bacterial growth and decreased biofilm formation	In vitro	[135]
*Bacillus* spp.	Isolate FI162, cells and cell-free supernatant, inhibited *T. maritimum* growth	In vitro	[143]
*Psychrobacter* genus, *Acinetobacter heamolyticus* and *Enterovibrio calviensis*	Antagonistic activity against *T. maritimum*	In vitro	[144]
*Psychrobacter nivimaris* and *P. faecalis*	Reduced fish mortality (rectal administrations)	*Psetta maxima*	[145]
*Phaeobacter piscinae* S26	The biofilm was effective in eliminating *T. maritimum*	In vitro	[146]
Marine actinomycetes	High antimicrobial activity against *T. maritimum*	In vitro	[147]
*Aspergillus niger β*-glucan MycoFence^®^ (Citribel, Tienen, Belgium)	Reduced fish mortality	*Salmo salar*	[141]

* NACs—natural antimicrobial compounds.

## 5. Theoretical Framework for Integration: Research Hypothesis and Validation Requirements

We present here our theoretical perspective on integrating thermal and nutritional strategies, emphasising that this represents our conceptual standpoint rather than empirically validated practice. To our knowledge, no published studies have tested combined thermal–nutritional protocols against *T. maritimum* or demonstrated synergistic effects in controlled trials. The integration concept should be understood as a research hypothesis requiring rigorous testing rather than an established management tool (Figure 4). Thermal and nutritional strategies operate through distinct biological pathways that could theoretically provide complementary benefits. Thermal strategies might provide acute immune enhancement during high-risk periods, while nutritional strategies could establish sustained baseline resistance through enhanced mucosal defence systems and beneficial microbiota composition.

Although combining different approaches could work well together in theory, there is also the possibility of conflicting interactions. For example, a temperature increase might enhance the efficiency of nutrient absorption, potentially improving the bioavailability of immunostimulant compounds delivered through functional feeds. Conversely, thermal stress may suppress appetite, impair digestive function, and disrupt gut microbiota composition, potentially diminishing the efficacy of interventions such as probiotics. The net impact of these interacting effects remains poorly understood and is likely influenced by species-specific physiology, environmental conditions, and the implementation of thermal intervention protocols. Accordingly, well-designed, species- and context-specific studies are essential to elucidate these dynamics.

Some nutritional studies offer indirect support for the concept. For instance, supplementation with *Ulva ohnoi* reduced *T. maritimum* colonisation in the gastrointestinal microbiota of fish [148], suggesting a foundation upon which thermal interventions might act during outbreak scenarios. However, extrapolating such findings to an integrated protocol is speculative without direct empirical validation. Timing is another challenge. Thermal treatments often need to be applied quickly during high-risk disease periods, while nutritional strategies usually require longer, consistent use to be effective. Aligning these two approaches is still a practical problem that requires further solution.

### 5.1. Species-Specific Considerations and Research Requirements

From our perspective, successful implementation would rely on precise species-specific calibration, taking into account thermal tolerance thresholds, dietary requirements, and immunophysiological traits. We present these considerations as research priorities that emerge from our analysis rather than established requirements. Warm-water species, such as gilthead seabream, might offer greater flexibility for thermal intervention strategies, with optimal culture temperatures of 18–26 °C providing theoretical room for controlled elevation to 28–29 °C without approaching the critical thermal maxima of 35.5 °C [45]. Conversely, cold-water species would require more conservative approaches due to narrow thermal tolerance ranges and different immune response kinetics. Accordingly, for species such as Atlantic salmon, integration strategies might highlight nutritional enhancement with limited thermal intervention. These species-specific differences underscore the need for comprehensive validation studies across focus species before any commercial recommendations can be made.

Prior to commercial implementation, the integration hypothesis requires rigorous empirical validation. This includes the following: (i) controlled factorial experiments to assess thermal and nutritional interactions with appropriate controls and statistical power; (ii) mechanistic studies elucidating molecular pathways linking thermal and nutritional responses; (iii) economic modelling grounded in real-world implementation costs rather than speculative projections; (iv) welfare assessment protocols to ensure interventions do not compromise fish wellbeing; and (v) commercial-scale trials across diverse production systems and environmental conditions.

While the integration of different interventions may appear conceptually reasonable, complex frameworks should be avoided during their validation. In practice, straightforward, well-validated single interventions could offer greater efficacy and cost-efficiency compared to multifaceted approaches that demand continuous monitoring and adjustments.

### 5.2. Technology Integration and Proteomic-Based Validation for Climate-Adaptive Management

Climate change is altering the ecological parameters in which aquaculture operates, notably by shifting thermal regimes, facilitating the spread of pathogens, and heightening the risk of disease emergence. These changes present significant challenges but also open opportunities for innovation through advanced technologies that enable proactive adaptation and mitigation to respond to this dynamic context.

Real-time temperature monitoring systems are essential in anticipating and managing temperature-related disease events [149]. Similarly, precision feeding technologies, particularly those employing AI-driven platforms, allow for dynamic nutritional adjustments based on continuous environmental and physiological inputs [150,151]. Furthermore, microbiome-based diagnostic tools are emerging as valuable instruments for the early detection of microbial imbalances and increased susceptibility to infections, supporting health management strategies in response to climate change [152].

Proteomics presents a promising tool for advancing welfare-focused approaches to disease management in aquaculture [153]. Analysing protein expression profiles may enable the identification of specific biomarkers that distinguish adaptive responses, such as behavioural fever, from stress-induced thermal responses, a differentiation that has remained challenging in thermal intervention research [154,155]. Key molecular indicators, including the expression of heat shock proteins (HSPs), immune-regulatory proteins, and stress-response molecules, may offer real-time insights into whether thermal strategies are supporting immune function or instead triggering physiological stress responses [156,157]. Assessing immune protein profiles, mucosal barrier integrity, and antimicrobial peptide production could yet provide standardised, mechanistic data on feed functionality, an area often limited by inconsistent methodologies [119,158,159].

Evaluating how thermal and nutritional strategies interact at the protein expression level may uncover synergistic or antagonistic mechanisms that remain speculative without molecular evidence.

Selective breeding programs targeting thermal tolerance are gaining momentum as a means of preparing fish populations for variable climate conditions [160]. These genetic improvements, when paired with tailored dietary interventions, have the potential to reinforce immune function and overall robustness [161]. Emerging evidence also points to a dynamic interaction between thermal regulation and immunonutrition, with possible downstream effects on gut microbiota composition and disease resistance [148]. To translate these insights into practice, technology development efforts should prioritise tools that enable precision and responsiveness. Key areas include automated systems capable of detecting behavioural fever, temperature control platforms tailored to aquaculture environments, and proteomics-based diagnostics for real-time welfare assessment. Additionally, integrated management systems that coordinate thermal and nutritional strategies are essential for optimising fish health in diverse production settings.

### 5.3. Sustainability Framework and Industry Transformation

While Section 5.2 explored the technological innovations supporting adaptive health strategies, these tools also underpin broader efforts to transform aquaculture into a more sustainable and welfare-driven food system. We propose that the combined use of thermal and nutritional strategies may offer a valuable opportunity to reshape aquaculture into a more sustainable, resilient, and welfare-focused food system. Rather than relying solely on reactive treatments like antibiotics, which are recognised to have long-term risks such as AMR, the focus should shift toward proactive immunity-based disease prevention. This shift is critical as AMR continues to threaten both aquatic animal health and public safety, prompting global calls to reduce antibiotic dependency across food production systems [73,162].

Integrating temperature modulation and functional feeds allows fish to activate their own natural defence mechanisms, supporting innate and mucosal immunity, controlling pathogen load, and promoting faster recovery when disease does occur. These non-pharmacological approaches can reduce the frequency and severity of disease outbreaks, thereby improving survival, reducing production losses, and minimising the need for chemical inputs [74]. Notably, and within the One Health framework, they also offer a way to deliver healthier products to consumers, with fewer residues and a lower environmental footprint. Nevertheless, to move beyond the proof-of-concept stage, broader industry transformation is needed. In this regard, proper collaboration among researchers, fish farm producers, nutritionists, technology developers, and regulators is crucial for developing standardised, scalable protocols that reflect the diversity of aquaculture systems and species. This includes testing how these integrative strategies perform in real aquaculture conditions, ensuring their cost-effectiveness, and facilitating knowledge transfer to farmers.

Climate change adds further urgency to this transition. As warming waters alter pathogen dynamics and stress aquatic species, robust disease prevention frameworks must become part of climate adaptation planning. Integrated approaches, if supported by data-driven tools such as real-time temperature monitoring, precision nutrition platforms, and welfare diagnostics, would help producers anticipate and respond to health risks before they escalate to high mortalities. Adopting these strategies extends beyond improved disease management; it represents a commitment to fostering a more ethical and sustainable aquaculture sector—one that prioritises fish welfare, safeguards public health, and maintains productivity in the face of environmental change.

## 6. Conclusions

This review critically evaluates thermal and nutritional strategies as welfare-oriented alternatives to conventional *T. maritimum* management, revealing significant theoretical potential constrained by substantial knowledge gaps and implementation challenges. Individual approaches demonstrate biological plausibility and preliminary efficacy in controlled settings. Thermal interventions offer acute immune enhancement potential through behavioural fever mechanisms and controlled heating protocols, particularly for warm-water species with adequate thermal tolerance margins. Nutritional strategies provide sustained immune support through functional feeds, probiotics, and immunostimulants, with broader species applicability but inconsistent commercial validation.

The integration hypothesis, combining thermal and nutritional approaches, represents our conceptual standpoint and currently lacks empirical validation in both controlled and commercial contexts. It should therefore be regarded as a research framework rather than an established management tool. However, rather than premature commercial implementation of unvalidated approaches, the aquaculture industry should invest in rigorous research programs that can responsibly advance these promising alternatives from theoretical concepts to practical management tools.

In this framework, a phased roadmap is essential. Initial research must focus on fundamental validation: factorial experiments with adequate statistical power, species-specific dose–response trials, and mechanistic studies using proteomics and microbiome analysis to differentiate adaptive responses from stress-induced pathologies. These must be followed by well-designed commercial-scale trials across diverse production settings, incorporating economic cost–benefit analyses and regulatory assessments. The next stage is a large-scale integration, supported by automated technologies for behavioural fever detection, precision feeding systems, and real-time welfare monitoring platforms. Success metrics for its validation should include significant reductions in antibiotic use, improvements in welfare indicators, and economic viability close to current practices.

As climate change introduces new and unpredictable stressors, the urgency for adaptive, proactive health frameworks intensifies. These debated welfare-oriented approaches are possible with coordinated efforts between researchers, producers, technology developers, and regulators. Public investment and industry collaboration are critical to overcoming logistical, regulatory, and knowledge-transfer barriers. Species such as gilthead seabream, prominent in Mediterranean aquaculture, could serve as model organisms for validating integration strategies, provided that trials are statistically robust and scalable.

From our perspective, realising the potential of integration will take more than technological innovation. We argue that it will require a fundamental shift in mindset across the sector, including adopting a culture of evidence-based decision-making, collaboration, and shared responsibility for animal welfare, environmental stewardship, and human health—in line with One Health principles. This represents our vision for the future of aquaculture disease management, offered to guide future research directions.

## Figures and Tables

**Figure 1 animals-15-02581-f001:**
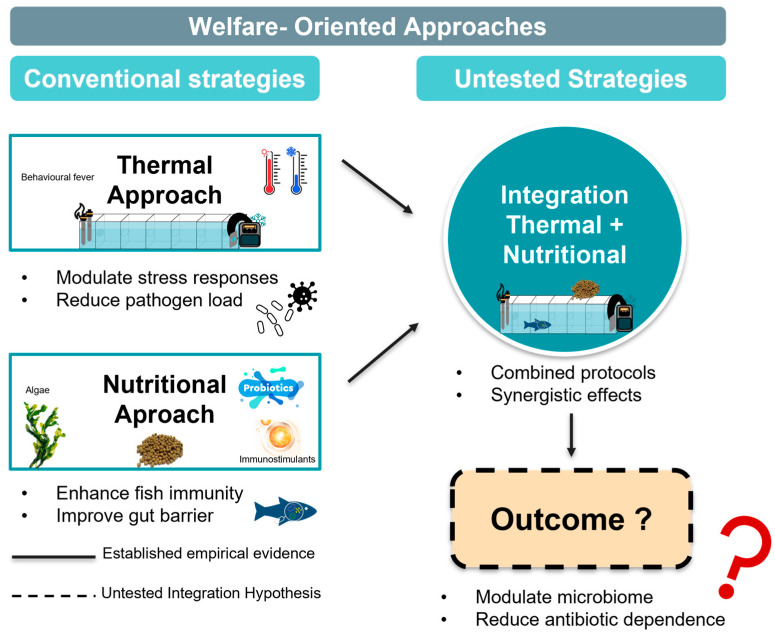
Conceptual framework distinguishing established evidence from untested integration hypothesis. Solid arrows represent pathways supported by empirical evidence; dashed arrows indicate proposed but unvalidated synergistic effects requiring future research validation.

**Figure 2 animals-15-02581-f002:**
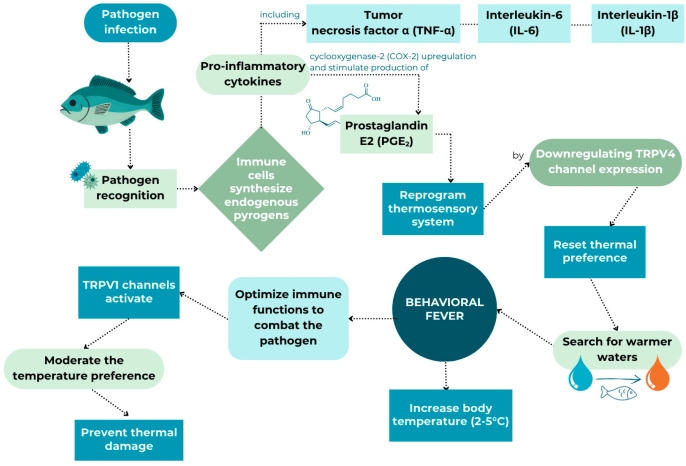
Proposed mechanism of behavioural fever in fish in response to infection, adapted from Boltana et al. [21,22].

**Figure 3 animals-15-02581-f003:**
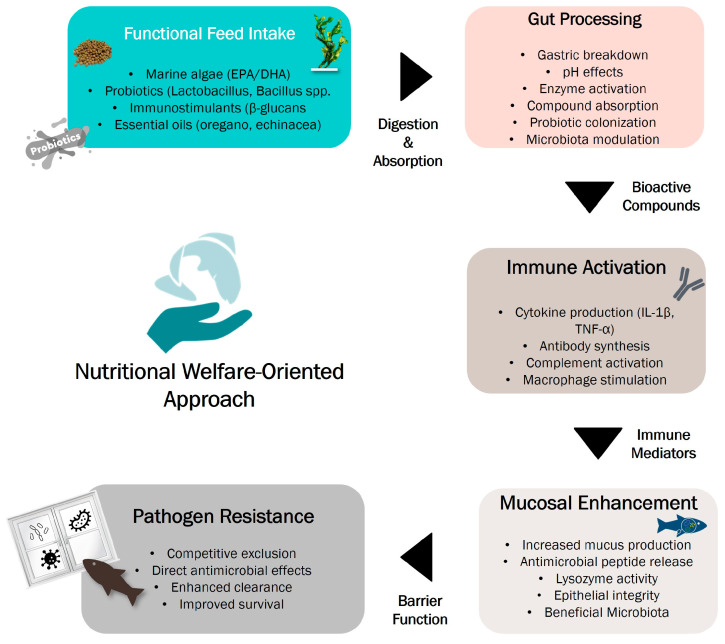
Nutritional pathway from functional feed intake to pathogen resistance. Functional feed components undergo gut processing and absorption, triggering immune system activation that enhances mucosal barrier function and provides multiple mechanisms of pathogen resistance. Arrows indicate sequential biological processes; boxes show key biological responses at each stage.

**Figure 4 animals-15-02581-f004:**
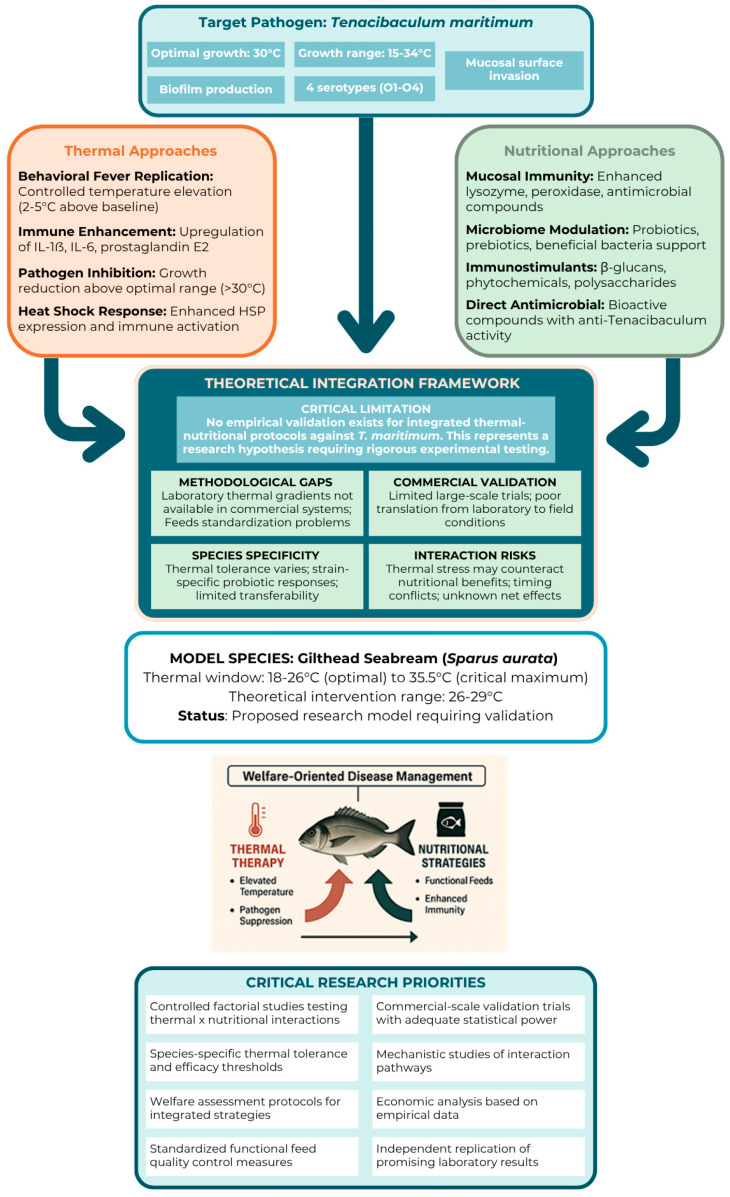
Proposal of welfare-oriented management framework for *Tenacibaculum maritimum* control in aquaculture. This illustration should be interpreted as a research hypothesis designed to guide future investigations rather than a validated management protocol.

**Table 1 animals-15-02581-t001:** Comparative analysis of conventional vs. welfare-oriented strategies for managing *Tenacibaculum maritimum* in aquaculture [19,63,73,74,75,76].

Aspect *	Conventional Treatments	Welfare-Oriented Alternatives	Scientific Rationale
Mechanism of Action	Direct pathogen suppression via antibiotics or chemicals	Support host resilience via immune modulation and environmental management	Avoids immunosuppression and reduces AMR risks
Stress Level	High (handling, confinement, Injections, or medicated baths)	Moderate to low (preventive, minimally invasive)	Reduced cortisol release and behavioural disruption
Effectiveness	Short-term; efficacy declines with resistance	Potentially long-term; still under commercial validation	Requires integrated, sustained application
Sustainability	Poor (antimicrobial residues, environmental impact, AMR emergence)	High (leverages natural defences, no chemical residues)	Aligns with One Health and welfare frameworks
Welfare Impact	Often negative (stress, handling injuries, welfare compromise)	Potentially positive (improved health, reduced handling)	Consistent with EU and OIE fish welfare recommendations
Empirical Support	Moderate to high (well-studied antibiotics; few species-specific vaccines)	Limited but growing (functional feeds, thermal protocols, probiotics)	Needs species- and system-specific trials to confirm efficacy and cost–benefit analysis

* Comparison based on peer-reviewed literature synthesis covering validated studies, welfare assessments, and sustainability analyses. Welfare impact ratings reflect consensus from fish welfare research; sustainability assessments incorporate environmental and economic factors; empirical support levels represent publication quantity and quality for each approach.

**Table 2 animals-15-02581-t002:** Thermal sensitivity of *Tenacibaculum maritimum* and implications for disease risk in selected species.

Species	Strain/Isolate	Thermal Range (°C) *	Infection Threshold	Peak Mortality/Outbreaks	References
*Dicologlossa cuneata*	a443 (EU623456)	~20.5 ± 1.5	Detected at ~20 °C	Increased incidence at ~20.5 °C	[91]
*Paralichthys olivaceus*	050603 and 46501	17–26	>17 °C	Peak mortality between 17 and 26 °C	[92]
*Dicentrarchus labrax*	N/A	18–28	>22 °C	More frequent outbreaks in summer	[54]
*Salmo salar*	89/4762	12–17	Induced with ECPs at 15–16 °C	Tissue necrosis observed	[93]
*Oncorhynchus mykiss*	Tm-035 and Tm-036	12–16	Outbreaks at 16.2 °C	Increased susceptibility >16 °C	[94]
*Sparus aurata*	N/A	18–26 (optimum)	Experimental data limited	Theoretical risk >28 °C	[45] Present review

* Temperature represents one of the multiple interacting factors influencing infection risk as discussed in Section 2.2. Listed temperatures should not be interpreted as definitive infection thresholds but rather as risk-associated ranges requiring integrated management approaches.

## Data Availability

No new data was created in this study, as it is a review of the existing literature.

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
