# Peer review of "Thermal and Nutritional Strategies for Managing Tenacibaculum maritimum in Aquaculture: A Welfare-Oriented Review"

_animals, 2025, doi:10.3390/ani15172581_

Round 1

Reviewer 1 Report

Comments and Suggestions for Authors

This is a well-organized, extensively referenced, and timely review that addresses an important and emerging topic in aquaculture: transitioning from pathogen-focused to welfare-oriented disease management. The manuscript is conceptually innovative and proposes integrated strategies that are scientifically plausible but not yet empirically validated. The use of gilthead seabream as a model species is appropriate and well-rationalized. However, the manuscript would benefit from enhanced clarity, more balanced critique, and careful attention to a few structural and content-related issues.

Author Response

We appreciate your thorough review and valuable suggestions for improving the manuscript's scientific rigor and critical analysis.

Comment 1: Central hypothesis integration untested - separate established findings from proposed hypothesis

Response: We agree this was a critical omission. We have added a new Section 1.1 "Methodological Approach" and we now clearly state that individual thermal and nutritional interventions have demonstrated efficacy in controlled studies, while their integration remains an untested hypothesis requiring rigorous validation. Additionally, we have added Figure 1 that visually distinguishes evidence-based strategies from hypothetical integration concepts.

Comment 2: Include detailed implementation barriers

Response: You are correct that practical constraints were underestimated. We have added a comprehensive new Section 2.5 "Implementation Barriers and Commercial Reality" that details such limitations.

Comment 3: Connect climate impacts to intervention design

Response: We have substantially revised Section 2.3 to demonstrate how climate change directly affects intervention strategy parameters. We now explain how rising baseline temperatures reduce thermal intervention windows and how intervention protocols must adapt to changing environmental conditions.

Comment 4: Merge Section 3.4 with Section 5 to avoid redundancy

Response: We have streamlined Section 3.4 to eliminate redundant content and cross-reference the comprehensive research framework in Section 5.

Comment 5: Adopt a more critical tone regarding limitations

Response: We have added substantial critical analysis throughout the manuscript. In the behavioral fever section, we now acknowledge that current understanding derives almost exclusively from laboratory studies that may not translate to commercial environments with multiple concurrent stressors. For functional feeds, we have added critical discussion of quality control failures, publication bias, and the gap between laboratory efficacy and commercial performance.

Comments 6-7: Simplify complex sentences and update literature

Response: We have simplified overly complex sentences throughout the manuscript. While we prioritized recent literature, we included foundational studies where: (1) recent research validates their continued relevance, (2) they established fundamental concepts still applicable today, or (3) recent studies on specific topics remain limited but needed to provide comprehensive coverage.

Reviewer 2 Report

Comments and Suggestions for Authors

The review article titled “Welfare-Oriented Approaches to Tenacibaculum maritimum Management in Aquaculture: A Critical Review of Thermal and Nutritional Strategies”, presents an appropriate and relevant discussion on alternatives to conventional antibiotic-based disease management in aquaculture. The focus of the paper on welfare-oriented approaches through temperature modulation and dietary interventions is praiseworthy given the increasing concerns about antimicrobial resistance and environmental impact.

Although the review is scientifically sound, it requires some major revisions before it is recommended for publication:

Although the abstract states it is a “narrative review”, a more systematic approach e.g., clearly defined search strategy, inclusion/exclusion criteria for studies, quantitative synthesis where possible, would significantly strengthen its scientific rigor and impact. Narrative reviews can sometimes be perceived as less objective or comprehensive than systematic reviews.

Although the abstract and conclusion mention “critical exploration” and “critical review”, some sections, particularly in the introduction and initial discussions of conventional methods, should benefit from a deeper critical analysis of the existing literature, perhaps by presenting conflicting views or details more clearly.

Even though behavioral fever is exciting, the review offers substantial room to it, in spite of recognizing the lack of direct evidence for T. maritimum and practical implementation challenges in large-scale aquaculture. This needs to be balanced with more emphasis on directly implementable thermal manipulation strategies.

The available text only briefly throws light on nutritional interventions in the abstract and simple summary. A critical review of this topic needs to require a dedicated and detailed section, similar to the one on thermal strategies, discussing specific ingredients (marine algae, probiotics, immunostimulants), their mechanisms, efficacy, limitations, and commercial validation. Without this, the review feels incomplete in addressing its stated scope.

Though the review correctly recognizes the lack of commercial authentication and knowledge gaps as limitations, it should suggest more concrete suggestions on how these gaps might be addressed in future research or industry collaborations.

The English language quality is generally good and clear. The text is easily understandable, and the scientific terminology has been used properly.

In conclusion, the review provides a valued contribution to the field but should be substantially supported by enhancing its methodological rigor, expanding the discussion on nutritional strategies, and deepening its critical analysis, alongside minor language refinements.

Comments on the Quality of English Language

The English language quality is generally good and clear. The text is easily understandable, and the scientific terminology has been used properly.

Author Response

Thank you for your suggestions to strengthen the methodological rigor and expand critical analysis.

Comment 1: More systematic approach needed

Response: While maintaining the narrative review format appropriate for our conceptual framework development, we have added Section 1.1 "Methodological Approach" that details our structured approach to literature identification, inclusion/exclusion criteria, and evidence synthesis methodology.

Comment 2: Deeper critical analysis needed

Response: We have substantially enhanced critical analysis throughout the manuscript, including new Sections: 2.5 "Implementation Barriers and Commercial Reality" and 3.2.1 “Limitations of Behavioural Fever" that examine methodological limitations in conventional treatment literature, publication bias, and inadequate welfare impact assessments. Overall, the different strategies were analyzed more deeply with increased critical assessment of the content, and additional information was added to address previous gaps.

Comment 3: Expand nutritional interventions discussion

Response: We have added comprehensive information that provides a detailed analysis of marine algae mechanisms, probiotic applications, strain specificity limitations, and aquaculture management practices.

Reviewer 3 Report

Comments and Suggestions for Authors

Dear Authors,

In this manuscript, the authors’ general approach is actually quite good. The consideration of welfare-oriented approaches for disease prevention is a valuable concept. However, upon reviewing the content of the manuscript, it becomes clear that the welfare status of fish species affected by T. maritimum is not fully specified. For the fish species in which this pathogen causes disease, the information provided is quite limited and refers only to the water temperatures in which the fish live. Unfortunately, considering only water temperature as an indicator of a fish’s well-being is an inadequate approach.

To ensure proper welfare in fish, many parameters must be considered beyond just water temperature, such as stocking density, cage depth, net mesh size, net structure, feeding frequency, water temperature, oxygen concentration and saturation, salinity, water current speed, and more. Therefore, if welfare is being considered in this manuscript’s approach, then for those fish species in which T. maritimum causes high mortality, these parameters should be examined and described in detail.

Apart from these parameters, the manuscript is also somewhat scattered in terms of content flow. What I refer to as “scattered flow” here is that the manuscript begins by addressing the limitations of conventional approaches before explaining the optimal parameters for the emergence of T. maritimum infection and the most susceptible fish species. I would have expected to first see information on which fish species exhibit high susceptibility to T. maritimum, under what conditions the infection develops, whether each strain shows the same mortality rate, or what specific characteristics strains with high pathogenicity have. Following this, I would have expected a discussion of conventional treatment methods and vaccines, which are known to be the most effective disease prevention strategies today. A comprehensive evaluation of all conventional methods reported in the literature, along with their advantages and disadvantages in terms of fish welfare, would have been very useful. Such a broad comparative overview would be highly beneficial for readers and would also increase the manuscript’s citation potential.

Furthermore, there is no mention of breeding disease-resistant strains. Therefore, the authors could consider including information on the cultivation of genotypes in these fish species that show resistance to T. maritimum.

In Table 2, the information provided suggests that infection might occur above the listed temperatures for each fish species. However, this inference is misleading if temperature is the only factor being considered. It is essential that other identified predisposing factors are also indicated and discussed. Otherwise, readers may come to believe that the disease will occur at those temperature values alone. This could lead to a false perception and should be carefully addressed.

Author Response

We appreciate your detailed feedback on welfare parameters and content organization.

Comments 1-2: Welfare parameters beyond temperature need specification.

Response: You are correct. We have added comprehensive information in section “2.2. Environmental and Management Drivers of Disease Emergence” that details how stocking density, water quality parameters (dissolved oxygen, ammonia, pH), feeding protocols, handling procedures, and social stress factors interact with temperature to influence infection risk. We also acknowledge these indicators' impacts throughout the paper where appropriate.

Comment 3: Reorganize content flow.

Response: We have improved the flow by restructuring section 2, swapping sections 2.2 and 2.3, adding a new subsection “2.5. Implementation Barriers and Commercial Reality,” and removing section 3.4 and topic 6 to eliminate redundancy.

Comment 4: Include breeding for disease resistance.

Response: We incorporated genetic approaches into Section 2.4, "Why Alternative Approaches Are Essential," with a brief discussion of heritability estimates, genomic selection potential, and how these can be integrated with thermal and nutritional strategies.

Comment 5: Clarify Table 2.

Response: We revised Table 2 and added a note below to clarify that temperature is one of multiple interacting factors. We reference Section 2.2 for a detailed discussion of other predisposing factors and clarify that the listed temperatures are risk-associated ranges, not absolute thresholds.

Reviewer 4 Report

Comments and Suggestions for Authors

Title

The title is too worded, I suggest it be changed to “Thermal and Nutritional Strategies for Managing Tenacibaculum maritimum in Aquaculture: A Welfare-Oriented Review”

Simple Summary

20-21: This phrases "significant challenge" and "substantial economic losses" are redundant because they present similar ideas

Abstract

 36: “by” was repeated in the sentence

40: This phrase “...could represent a paradigm shift...” appears too ambitious for  review that empirical validation was not employed

45: Be specific on how “Transitioning to welfare-oriented aquaculture requires rigorous evaluation and validation…”

Keywords

Some keywords like “temperature” is vague, they should be replaced with precise terms

Introduction

51–52 and 66: Check sentence structure

56: “...rising temperatures, shifting precipitation patterns and ocean acidification…” is listed without citation relevance.

84–85: Why is short duration immunity conferred by “ICTHIOVAC-TM problematic?

Limitations of Conventional Approaches

104-105:  Serotypes (O1–O4) was mentioned but I expected to see molecular typing challenges

142: Give specific examples of “...conflict with long-term sustainability...”

148: There may be need to reorganize the contents, with AMR introduced earlier

150-152: In which region are amphenicols and fluoroquinolones used commonly in aquaculture?

Table 1: Add references and a footnote indicating basis for comparison  

Thermal Strategies

294: Please clarify “...stress-induced ‘emotional fever’ is complex…”

321: These variables should be linked to pathogen susceptibility “...variable oxygen levels, social interactions...”

Nutritional Strategies

465: This important statement appears to be buried “Because T. maritimum often enters the host through mucosal surfaces…”

475–481: What is the evidence that Phytochemicals for immunostimulatory are “Widely investigated” and “inconsistent outcomes”

520: What specific standards are lacking ? “...still awaits standardization of functional ingredient quality...”

A graphical representation such as flowchart showing the pathway of diet to mucosal immunity to resistance would facilitate a reader’s understanding

Integration Framework

574: Is this a disclaimer? “Untested Hypothesis Requiring Validation”

 592–596: This is contradictory and speculative, is there reference to back it up?

636–673: Lots of ideas

Case Study: Gilthead Seabream

740: This is repetitive “...to our knowledge no studies have tested thermal intervention…”

747: What is the functional consequence of this “...reduced it” (microbial diversity)

750: Italicize name of organism and throughout the manuscript

Conclusion

The conclusion is well summarized, but authors should specify whether experimental design, cost-benefit studies, field validation, etc is required for future research. Also, they should conclude as a scientific roadmap rather than a general call for action

Author Response

Thank you for your detailed, section-specific feedback that has improved the manuscript's clarity and precision.

Comments 1-6: Title, Simple Summary, Abstract, Keywords revisions

Response: We have implemented all suggested changes: shortened the title, eliminated redundant phrases, corrected repeated words, toned down ambitious language about "paradigm shifts," specified validation requirements, and replaced vague keywords with precise terms.

Comments 7-9: Introduction improvements

Response: We have corrected sentence structure issues, added citation relevance for climate change effects, and explained why short-term vaccine immunity is problematic.

Comments 10-14: Conventional approaches section

Response: We have added discussion of molecular typing challenges beyond serological classification, provided specific examples of sustainability conflicts, reorganized AMR content for better flow, specified regional patterns of antibiotic use, and enhanced Table 1 with references and explanatory footnote.

Comments 15-16: Thermal strategies clarifications

Response: We have clarified the complexity of distinguishing beneficial behavioral fever from stress-induced responses and linked environmental variables directly to pathogen susceptibility and behavioral fever suppression. We have added a new subsection “3.2.1 Limitations of Behavioural Fever” to better organize the ideas behind this concept.

Comments 17-20: Nutritional strategies improvements

Response: We have highlighted the important statement about mucosal surface entry, provided evidence for "widely investigated" claims with publication numbers, specified lacking standardization requirements, and added Figure 3 showing the nutritional pathway from diet to mucosal immunity to resistance.

Comments 21-26: Integration framework and case study

Response: We have clarified the section title rationale, addressed contradictory content with supporting references, condensed the lengthy technology discussion, removed repetitive statements, explained functional consequences of microbial diversity changes, and italicized all organism names throughout.

Comment 27: Scientific roadmap conclusion

Response: We have completely revised the conclusion to provide a specific scientific roadmap with three research phases (Fundamental Validation, Commercial Translation, Industry Integration) and success metrics.

Round 2

Reviewer 1 Report

Comments and Suggestions for Authors

Good revision.

Author Response

Thank you for your feedback on our revised manuscript. We appreciate your acknowledgment that this version represents a significant improvement over the initial submission.

In response to your recommendation and Reviewer 3's comments, we have carefully revised the manuscript to explicitly frame our work as our scientific perspective intended to encourage discussion rather than present a fully validated endpoint model.

Key revisions:

  1. We have clarified our integration hypothesis as our theoretical framework, which requires empirical validation rather than being an established approach.
  2. We have adjusted language throughout the manuscript to clearly indicate that this represents our perspective and interpretation of the available evidence.
  3. We have highlighted the discussion-stimulating nature of the review, especially in the abstract, introduction, and conclusions.
  4. We have strengthened some sentences about validation requirements and research priorities that should be addressed before practical implementation.
  5. We have clarified in the methodological section that our narrative approach inherently reflects our scientific perspective.

These changes preserve the scientific rigor and valuable contributions of our analysis while properly positioning the work as a thoughtful perspective aimed at advancing the field through research and scientific discussion rather than definitive conclusions.

We believe these revisions address the reviewer concerns directly while maintaining the manuscript’s contribution to promoting welfare-oriented approaches in aquaculture disease management.

Thank you for your guidance throughout this process. We look forward to your final evaluation.

Sincerely,

Raquel Carrilho, Márcio Moreira, Ana Paula Farinha, Denise Schrama, Florbela Soares, Pedro Rodrigues, Marco Cerqueira

Reviewer 2 Report

Comments and Suggestions for Authors

The updated manuscript has been subjected to an additional cycle of rigorous peer review, wherein the modifications implemented in response to the prior review were systematically evaluated.

In their resubmission, the authors have undertaken substantial revisions to comprehensively address the critiques outlined in the initial review. Furthermore, they have furnished well-substantiated responses to the specific inquiries raised during the evaluation process.

By systematically addressing reviewer feedback and integrating the recommended revisions, the manuscript has undergone significant refinement, resulting in marked improvements in clarity, rigor, and overall quality. The only persisting deficiencies are minor linguistic inaccuracies, which can be efficiently resolved during the final proofreading stage prior to publication, contingent upon acceptance.

Author Response

(The authors gave the same response as above.)

Reviewer 3 Report

Comments and Suggestions for Authors

Dear Authors,

While the revised manuscript demonstrates some improvements, several critical limitations previously highlighted remain insufficiently addressed.

Author Response

(The authors gave the same response as above.)
